# Bacteriophage Therapy as an Application for Bacterial Infection in China

**DOI:** 10.3390/antibiotics12020417

**Published:** 2023-02-20

**Authors:** Shuang Liang, Yanling Qi, Huabo Yu, Wuwen Sun, Sayed Haidar Abbas Raza, Nada Alkhorayef, Samia S. Alkhalil, Essam Eldin Abdelhady Salama, Lei Zhang

**Affiliations:** 1College of Animal Science and Technology, Jilin Agricultural University, Changchun 130000, China; 2Borui Technology Co., Ltd., Changchun 130000, China; 3College of Animal Science and Technology, Northwest A&F University, Yangling 712100, China; 4Department of Clinical Laboratory Science, College of Applied Medical Sciences, Al-Quway’iyah, Shaqra University, Riyadh 19257, Saudi Arabia; 5Department of Clinical Laboratory Sciences, Faculty of Applied Medical Sciences, Shaqra University, Shaqra 11961, Saudi Arabia; 6Anatomy Department College of Medicine, King Saud University, Riyadh 11495, Saudi Arabia

**Keywords:** phage therapy, animal models, clinical application, anti-infection, bacterial resistance

## Abstract

Antibiotic resistance has emerged as a significant issue to be resolved around the world. Bacteriophage (phage), in contrast to antibiotics, can only kill the target bacteria with no adverse effect on the normal bacterial flora. In this review, we described the biological characteristics of phage, and summarized the phage application in China, including in mammals, ovipara, aquatilia, and human clinical treatment. The data showed that phage had a good therapeutic effect on drug-resistant bacteria in veterinary fields, as well as in the clinical treatment of humans. However, we need to take more consideration of the narrow lysis spectrum, the immune response, the issues of storage, and the pharmacokinetics of phages. Due to the particularity of bacteriophage as a bacterial virus, there is no unified standard or regulation for the use of bacteriophage in the world at present, which hinders the application of bacteriophage as a substitute for antibiotic biological products. We aimed to highlight the rapidly advancing field of phage therapy as well as the challenges that China faces in reducing its reliance on antibiotics.

## 1. Introduction

At present, the main treatment for bacterial infection is antibiotics. Due to the extensive use of antibiotics, there have been numerous problems, such as the emergence of drug-resistant bacteria, immunosuppression, drug residues in animal products, environmental pollution, and so on. According to statistics, 75% of bacteria in the United States are resistant to at least one kind of antibiotic, and nearly 2 million people’s health is at risk each year due to drug-resistant bacteria [1]. In addition, more than half of the clinical *Staphylococcus* isolated in Japan have multidrug resistances to antibiotics [2]. A report from the British government in 2016 showed that the deaths caused by drug-resistant bacteria could reach about 700,000 each year [3]. It is estimated that by the year 2050, upwards of 10 million people will die each year due to antimicrobial resistance. In 2017, the World Health Organization (WHO) published a list of global priority pathogens comprising 12 species of bacteria which were categorized into critical, high, and medium priority based on their level of resistance and available therapeutics (Table 1). It is crucial to discover, design, and develop new and alternative antimicrobial therapies. The current rate of resistance development far exceeds the level of antibiotic discovery and development and represents a global public health challenge.

The Chinese government placed a high priority on bacterial drug resistance and included it on the agenda of the G20 Hangzhou Summit. In order to control the development of bacterial resistance, the Ministry of Agriculture and Rural Affairs of the People’s Republic of China issued *the National Action Plan for Reducing the Use of Veterinary Antimicrobials (2021–2025)* and *the List of Banned Drugs and Other Compounds for Food Animals*. In addition, the Chinese government encourages large-scale animal husbandry to cut off pathogen infection at the source. Companies have tried to develop new antibiotics, but few are available in terms of their commercial value [4]. Faced with the problem of increasing bacterial resistance year by year, it is urgent to find a new treatment that can replace antibiotics. Antimicrobial peptides and biological enzymes are also popular alternatives to antibiotic therapy. However, the utilization of these medications is constrained due to their lengthy research cycles and limited antibacterial properties [5].

Bacteriophage(phage) is a virus that can infect microorganisms such as bacteria, fungi, actinomycetes, and spirochetes. Phages have been used as antibacterial agents since their discovery in the 1920s. However, with the discovery of antibiotics and their widespread use, people gradually ignored the in-depth study of phage therapy. Since the 1980s, due to the continuous emergence of drug-resistant bacteria worldwide, antibiotic therapy has been facing great challenges, and bacteriophage as a traditional antimicrobial therapy has attracted attention again. Phage has the characteristics of strong specificity, little toxicity, good bactericidal effect, good biological safety, and great potential in the prevention and control of bacterial infection. Recently, some scholars from western countries have published special papers on bacteriophage antibacterial technology, sharing positive comments on the basic research and development of bacteriophage. They believed that the natural targeting effect, specificity, and high efficiency of bacteriophages had opened up a new field for the control and prevention of bacterial diseases [6,7,8]. Haddad et al. collated the literature on phage therapy published from 1985 to 2018, and the results showed that phage therapy was effective in reducing bacterial concentration, degrading biofilms, healing wounds, and improving outcomes in most studies (87%, 26/30) [9].

In this review, we summarized the research and application of phage in the prevention and control of bacterial infections in China, and discussed the limitations and challenges of phage therapy, hoping to provide a reference for promoting the clinical application of phage therapy.

## 2. Overview of Phages

Bacteriophages (phages) are highly abundant in the environment and may be the source of low-cost antimicrobials. Bacteriophages coexist with their bacterial hosts and play an important role in many biological processes such as bacterial evolution and microorganism diversity [10,11,12,13,14,15]. In general, the biological characteristics of a phage include its optimum pH, temperature, one-step growth curve, morphology, etc. Like other viruses, phages are simple in structure, consisting of a protein capsid and core. According to their morphological and structural characteristics, there are three types of phages: *Caudovirales* (which are divided into *Siphoviridae*, *Podoviridae*, and *Myoviridae*), *Ballabactivirus*, and *Inoviridae* [16]. Ling Chen et al. described the biological characteristics of phage ValSw3-3, which belonged to the *Siphoviridae* [17]. ValSw3-3 had a latent period of about 15 min and a burst size of 95 ± 2 PFU/cell. Its infectivity remained above 80% at pH 6–10. Meanwhile, this phage was able to exist stably at 4–50 °C. Because of their tail fiber proteins, which can specifically recognize receptors on the surface of bacteria, phages can form a strict specificity with the host bacteria. Although the life cycle of phages is well understood, we will discuss it briefly based on the article of Zhang (Figure 1) [18]. According to the life cycle of phages, they can be divided into lytic phages and temperate phages. Lytic phages, also known as virulent phages, have the ability to inject their genomes into the bacteria, hijack the metabolic function of the host, and lyse the host cells to produce new progeny phages [19]. Temperate phages live a different life cycle and infect their hosts by initiating a lysogenic cycle. In the lysogenic cycle, the phage genome remains dormant as a prophage, replicates alongside its host, and occasionally bursts into a lytic cycle in response to a specific trigger [20].

## 3. Phage Therapy in Animal Models

By 2022, we have found more than 95,400 articles about phage research on the National Center for Biotechnology Information (NCBI) database (URL: https://www.ncbi.nlm.nih.gov/, accessed on 27 July 2022), of which more than 4500 were published by Chinese researchers, accounting for 4.72% (Figure 2). Along with the gradual deepening of phage research, scientists have assessed the value of phages in animal models through more rigorous and detailed experiments. We are attempting to explore the possible clinical application of phage therapy based on the animal experiments. We summarized the progress of phage application in mammals, oviparous animals, and aquatic animals by Chinese researchers (Table 2).

### 3.1. Mammals

Many studies have shown that bacteriophages have a good therapeutic effect in mammals. Mice models are widely used in the treatment of diseases such as pneumonia, sepsis, and intestinal infection with phage therapy. As a result of a study conducted by Cao, after mice were infected with multiple drug-resistant *Klebsiella pneumoniae* bacteria for 2 h, the survival rate could reach about 80% by nasally inhaling phage 1513 (2 × 10^9^ PFU/mouse) [37]. Moreover, the injury of the lung was effectively improved in the treatment group, while the mice in the untreated control group all died, indicating that phage had a good effect on lung infection caused by drug-resistant bacteria. Wang infected mice with imipenem-resistant *Pseudomonas aeruginosa* [42]. In this experiment, all the animals died within 24 h after the minimal lethal dose of bacteria (3 × 10^7^ CFU/mL) injected. If phage was injected 15 and 30 min after the bacteria infection (MOI > 0.1), the survival rate could reach 100%. In contrast, the survival rate decreased to 50% and 20% respectively, when phage treatment was administered at 3 h and 6 h. Thus, phages should be used as early as possible to treat bacterial infections. Bacteriophages are also effective in treating *Escherichia coli*, *Acinetobacter baumannii,* and *Yersinia enterocolitica* infections [49,50,51]. *Yersinia enterocolitica* is generally considered an important food-borne pathogen, particularly in the European Union. Xue established a mouse enteritis model caused by a *Yersinia enterocolitica* infection with serotype O:3, and evaluated the therapeutic effect of the instillation of phage X1 [47]. The result showed that a single oral administration of phage X1 (1.95 × 10^8^ PFU/mouse) at 6 h post infection was sufficient to eliminate *Yersinia enterocolitica* in 33.3% of mice (15/45). In addition, the number of *Yersinia enterocolitica* strains in the mice was also dramatically reduced to approximately 10^3^ CFU/g after 18 h, compared to 10^7^ CFU/g in the mice without phage treatment. Phage X1 treatment significantly improved of intestinal tissue, and the level of pro-inflammatory cytokines (IL-6, TNF-α and IL-1β) was significantly reduced (*p* < 0.05). These results indicated that phage was a promising candidate to control infection by bacteria in mice. In mice models, Prof. Hongping Wei and Prof. Hang Yang from the Laboratory of Diagnostic Microbiology, CAS Key Laboratory of Special Pathogens and Biosafety, Center for Biosafety Mega-Science, Wuhan Institute of Virology, Chinese Academy of Sciences also contributed to the application of phage and lysin therapy in the treatment of drug-resistant bacterial infections [52,53].

Phage therapy is also widely used in large mammals such as pigs, cattle, and sheep. For weaned piglets, *Escherichia coli* is the main pathogen causing diarrhea. In 2019, Zeng YD et al. found that adding phage in diets can improve growth performance and reduce the diarrhea index of weaned piglets [54]. Moreover, the phage supplement can improve the morphological structure and function of intestines, regulate the activity of intestinal digestive enzymes, improve the structure of intestinal microorganisms, and promote the intestinal health of piglets [55]. Foot-and-mouth disease (FMD) is a highly contagious disease in cloven-hoofed animals, which causes severe economic losses. To prevent this disease, Hai demonstrated the potential of T7 bacteriophage, based on nanoparticles displaying a genetically fused G-H loop peptide (T7-GH) as a FMDV vaccine candidate [56]. They found that the T7-GH phage nanoparticles were effective in eliciting antigen-specific immune responses in pigs. Qu YG invented a bacteriophage cocktail (six species of *Escherichia coli* phages of bovine mastitis; two species of cow mastitis source *Streptococcus* phages; two dairy cow mastitis, source *Staphylococcus aureus* bacteriophage; 1 × 10^9^ PFU of each phage) in 2021 for clinical dairy cow mastitis and recessive mastitis, of which the effect was superior to antibiotic treatment and antimicrobial peptide treatment [57,58]. There were many other experiments on the successful treatment of bacterial infection in mammals by phage, but we have not provided any more examples.

### 3.2. Ovipara

Phage can be used as a prophylactic agent to protect chickens from lethal *Escherichia coli* infestation. Inoculation of the phage (BP16, 1.5 × 10^8^ PFU) suspension prior to infection with *Escherichia coli* (O2 serotype, 1.5 × 10^8^ CFU) enabled a 100% protection rate [59]. When the chicken were injected with phage after the bacteria infected them, the survival rate could reach around 80%, compared with 70% in the antibiotic-treated group. The results showed that phage was effective in preventing and treating poultry infection caused by lethal *Escherichia coli*. *Salmonella pullorum* is the major pathogen that is harmful to the poultry industry in developing countries, and the treatment of chicken diarrhea caused by drug-resistant *Salmonella pullorum* has become increasingly difficult. A lytic bacteriophage of YSP2 was used, which was able to specifically infect *Salmonella* [32]. Experiments in vivo demonstrated that a single oral administration of YSP2 (1 × 10^10^ PFU/mL, 80 μL/chicken) 2 h after *Salmonella pullorum* administration at a double median lethal dose was sufficient to protect chickens against diarrhea. Bacteriophages and their phage cocktails are also widely used as disinfectants specifically to eliminate pathogens in poultry [60,61].

### 3.3. Aquatilia

People often use antibiotics and water disinfectants in aquaculture to treat bacterial infections. However, the entry of residual drugs into the natural environment affects the composition and activity of microorganisms, disrupting the balance of microbiome. Phage therapy is used in aquaculture, which can effectively control pathogenic bacteria including *Citrobacter freundii*, *Aeromonas hydrophila*, *Vibrio parahaemolyticus*, *Vibrio harveyi*, etc. Zhang Lei et al. from Jilin Agricultural University, committed to the research of phage therapy, found that bacteriophage had good therapeutic effects in the treatment of aquatic drug-resistant bacteria. They isolated a phage, IME-JL18, which had strong activity against *Citrobacter freundii*, and applied it to the treatment of carp enteritis models [43]. In this study, a dose of IME-JL18 at 1 × 10^7^ PFU effectively counteracted the lethal dose of *Citrobacter freundii* (1 × 10^9^ CFU/carp), inhibiting the formation of the host bacterial biofilms. To treat an *Aeromonas hydrophila* infection, a phage mixture therapy was established based on the analysis of the genomic sequences and biological characteristics of vB_AHAp_PZL-AH8 and vB_AHAp_PZL-AH1 [62]. The results also showed that phage therapy was a good method to inhibit the production of phage-resistant strains. This team recently found a bacteriophage named PZL-Ah152, which had good efficacy in reducing the pathogenic *Aeromonas hydrophila* strain 152 in vivo and in vitro [63]. Furthermore, a 12-day consecutive injection of PZL-Ah152 (2 × 10^9^ PFU) did not cause significant adverse effects on the main organs of the treated fish, such as liver, spleen, kidney, gut, and gill. They also found that members of the genus *Aeromonas* can enter and colonize the gut. The phage PZL-Ah152 reduced the number of colonies of the genus *Aeromonas*. However, no significant changes were observed in α-diversity and β-diversity parameters, which suggested that the consumed phage had little effect on the gut microbiota. Other researchers have also studied the application of bacteriophages in aquatic products. A study found that vB_VpaS_PG07 (7.6 × 10^9^ PFU/mL) significantly reduced the mortality of shrimps challenged with *Vibrio parahaemolyticus* (2.27 × 10^6^ CFU/shrimp), a bacterium causing acute hepatopancreatic necrosis disease (AHPND). The findings highlighted the potential of PG07 as an effective antibacterial agent for phage prophylaxis and phage therapy in aquaculture [64]. Another study found that phage qdvp001 (1.0 × 10^7^ PFU/mL) could purify *Vibrio parahaemolyticus* (1.0 × 10^8^ CFU/mL) in both oyster cultured environment and oyster bodies [65]. Simultaneously, this phage could inhibit the expression of pro-inflammatory factors including IL-1β, IL-6, and CD-14, and regulate the immune response caused by the bacteria. In addition, *Vibrio harveyi* can cause infections and diseases in a variety of marine vertebrates and invertebrates, which is very harmful to the aquaculture industry. Cui et al. isolated two bacteriophages [66], vB_VhaP_Vh-5 and vB_VhaM_VH-8, and discussed the practicability of feeding phages as a route of administration to protect turbot from *Vibrio harveyi* infection. When the MOI was 10 and 100, the phages could enhance the resistance of turbot to *Vibrio harveyi* VH5 infection, indicating feeding phage cocktails may be another optimal therapeutic agent against *Vibrio harveyi* infection in turbot culture.

## 4. Application of Bacteriophages in Clinics

In the 1940s, with the discovery of Penicillin, rapid and effective sterilization became the most effective treatment for bacterial diseases, heralding the golden age of antibiotics. Scientists have largely abandoned in-depth research on phage therapy ever since [67]. However, scientists in the former Soviet Union and some eastern European countries were not affected by the abandonment of phage therapy by western European countries and the United States, and still carried out in-depth research on the anti-infection effect of phages [68,69]. Currently, phage preparations have been approved for commercial use in many countries, including Georgia, Poland, and Russia. In China, the application of phage therapy to treat clinical diseases also has a long history. The former Dalian Institute of Biological Products was the first to develop phage research and related production (used for the prevention and treatment of dysentery) in China [70]. The Wuhan Institute of Biological Products also carried out a period of trial production around 1958. In 2017, the Shanghai Institute of Phage was established, which was the first institution to obtain the qualification for clinical treatment of bacteriophages in China. The institute launched the first ethically approved clinical trial of phage therapy in 2018 in China. Since then, teams such as the Chinese Academy of Sciences have also been working on the development of engineered phages to treat drug-resistant bacterial infections. In 2020, the Chinese team published a paper on the clinical practice of bacteriophages and proposed a new protocol of “non-sensitive antibiotic-phage combination,” providing a new strategy for the treatment of “super-bacteria” that was resistant to both antibiotics and bacteriophages [71]. The leading enterprises of phage industrialization in China also include Qingdao Nuoan Baxter Biotechnology Co., Ltd. and Phagelux Inc. These companies applied bacteriophage formulations to animals, aquatic products, agriculture, and human health. Here, we summarize some successful cases of phage application in clinical therapy in China (Figure 3), hoping to promote the wider application of phage in clinical practice (Table 3).

### 4.1. Skin Infection

Gram-negative bacilli, including *Escherichia coli*, *Klebsiella pneumoniae*, *Pseudomonas aeruginosa*, and *Acinetobacter baumannii*, are important pathogens of nosocomial infections. The majority of patients who suffered from chronic wounds/ulcers (86.1%, n = 310) and skin infections (94.9%, n = 734) experienced remission or improvement after phage therapy [78]. In 1958, Professor Yu He successfully cured patients infected with *Pseudomonas aeruginosa* using phage therapy [73]. This was the first successful clinical application of phage in China. In 2021, a patient in Shanghai Public Health Clinical Center was infected with drug-resistant *Pseudomonas aeruginosa* after surgery. Following expert consultation, phage therapy with catheter perfusion and wound wet compress was implemented. After the treatment, the patient was cured.

### 4.2. Respiratory Tract Infection

Clinically, aerosol inhalation techniques enable accurate delivery of bacteriophages to the site of infection [79]. In the treatment of lung infection caused by drug-resistant *Klebsiella pneumoniae*, clinical symptoms were effectively relieved after two inhalations of phage aerosol preparations [80]. The team led by Ma also used the aerosol method of bacteriophage to treat patients and, after two weeks of treatment, successfully cleared the lung infection caused by *Acinetobacter baumannii* [72]. This was the first case in Shenzhen (China) using phage to treat drug-resistant bacteria infections. Furthermore, Wu assessed the efficacy and safety of compassionate phage therapy on secondary carbapenem-resistant *Acinetobacter baumannii* (CRAB) infections in patients hospitalized with critical COVID-19. They suggested the potential of phages on rapid responses to secondary CRAB outbreak in COVID-19 patients [81,82]. For the patient who suffered from chronic obstructive pulmonary disease (caused by carbapenem-resistant *Acinetobacter baumannii*), Tan used a specific lysing pathogen-specific phage (phage Ab_SZ3, 5 × 10^6^ PFU-5 × 10^10^ PFU) in combination with tigecycline and colistin for a 16-day treatment [83]. The patient’s pathogen clearance rate and lung function were clinically improved.

### 4.3. Urinary Tract Infection

Urinary tract infection (UTI) with extensively drug-resistant *Klebsiella pneumoniae* (XDRKp) is a challenging infection complication to immunocompromised patients, such as transplant recipients and patients with cancer and diabetes. In the context of increasing antibiotic resistance, phage therapy is effective in treating urinary tract infections. In 2018, Zhu successfully cured the first urinary tract infection caused by multidrug-resistant *Klebsiella pneumoniae* in China. In 2019, another patient infected with complex pan-drug-resistant *Klebsiella pneumoniae* recovered by using a 3-strain phage mixture through continuous bladder perfusion and retrograde ureteral intubation in both renal–pelvis [76]. The following year, Bao reported a case of a 63-year-old female patient who developed a recurrent urinary tract infection with extensively drug-resistant *Klebsiella pneumoniae* (ERKp) [71]. The combination of antibiotic (Trimethoprim-sulfamethoxazole, SMZ-TMP, 800mg-100mg) with a phage mixture (Kp152, Kp154, Kp155, Kp164, Kp6377, and HD001, 5 × 10^8^ PFU/mL for each phage) inhibited the appearance of phage-resistant mutants in vitro and successfully cured the patient. There is a possibility that, instead of replacing antibiotics with phages, a combination of these two types of antibacterial agents may be more effective compared with use of either independently [84]. Qin et al. used a cocktail phage therapy to treat a man with multifocal urinary tract infection caused by MDR *Klebsiella pneumoniae* in 2021 [85]. Finally, the patient recovered. These successful cases provided valuable treatment ideas and solutions for phage treatment of complex infections.

### 4.4. Others

In addition to the above aspects, Chinese researchers are also attempting to apply phages in the treatment of chronic rhinosinusitis, dental ulcers, and alcoholic hepatitis. In 2018, Zhang identified bacteriophages (Sa83 and Sa87, 1 × 10^5^ PFU) that could effectively kill multidrug-resistant *Staphylococcus aureus* in the nasal passages of patients with chronic sinusitis [86]. Moreover, Li used phage topical application to treat the tuberculous oral ulcer, and the patient soon recovered [74]. Meng found that using *Enterobacter faecalis* phage can significantly reduce liver cytolytic level and fecal enterococcus quantity [75], which can eliminate ethanol-induced liver injury and steatosis. Phages can target specific bacteria, providing a new treatment for alcoholic hepatitis.

## 5. Limitation of Phage Therapy

There is no doubt that phage therapy is an attractive solution to combat escalating antibiotic resistance. Numerous studies have highlighted the potential of therapeutic phages in vitro and in vivo, and many clinical trials have been conducted over the past decade [87]. Despite the promise of phage therapy, many issues must be addressed before it can be used as a treatment: ① The immune response. As a foreign substance, phage can stimulate the immune response of the body, and the antibodies produced may inhibit the ability of phage to eliminate bacteria, thereby reducing phage efficacy [88]. Moreover, the reticuloendothelial system can quickly eliminate phages. Optimizing the dose of phage can reduce the production of specific antibodies. In addition to reducing the dose of phage application, phage packaging can also alleviate specific antibodies [89]. Similarly, an encapsulation approach for standard phage therapy can also provide effective and targeted delivery of phages to the site of infection to protect the phages from immune system clearance [90]. ② The host spectrum of phages is relatively narrow; a certain type of phage can only infect and lyse the corresponding species or type of bacteria [91]. Phage cocktail therapy can combine species–specific phages to improve the efficacy of phage therapy and also reduce the production of phage-resistant mutants [92]. ③ Phage therapy has limitations in dose and duration. According to the pharmacokinetics, phages can only proliferate when the bacteria reach a certain density [93]. Therefore, early or inappropriate bacteriophage inoculation may be cleared by the immune system before proliferation. Optimal inoculation time and dose will be the key to phage therapy. ④ When bacteriophages are stored for a long time, there will be a certain inactivation or reduced titers. Bacteriophages as viruses have protein structures, so they are susceptible to known protein denaturation factors. These factors include exposure to organic solvents, high temperatures, pH, ionic strength, and interface effects [94]. Therefore, when choosing a storage method, careful consideration of its physical and chemical properties is required. The current preservation methods are: a. Broth lysate; b. 50% glycerol dilution; c. Saturating filter paper with lysate and then drying; d. Phage lyophilization [95]. ⑤ Safety of phage application. Some toxin genes carried by phages may have adverse effects on the body. Therefore, phages should be sequenced before they are used to ensure that they do not have virulence genes or integrase genes. The important step in the process of isolation and purification of phage is to remove endotoxin. Bacterial endotoxin, as an important part of Gram-negative bacteria, is released when bacteriophage lyses bacteria, which is toxic to the human body and has potential safety hazards. At present, bacteriophage preparation can be purified by affinity chromatography, commercial kits, and other methods to remove endotoxin from bacteriophage particles. ⑥ Criteria for clinical use of phages. It is necessary to ensure the activity of phages and remove contaminants from bacteria as much as possible. Information critical to the success of clinical trials includes: a. The adequate characterization and selection of phages as well as of subjects (humans) and the target bacteria. b. The formulations, dosing, and efficacy of the phage [96]. c. The sterility and purity of the phage preparation [97].

## 6. Regulation of Phage Application in China and Western Countries

Phage therapy has made great achievements both in animals and clinically. When it comes to the clinical treatment of phages, we definitely mention Hirszfeld Institute in Wroclaw, Poland and the Eliava Institute in Tbilisi, Georgia. The Eliava Institute focuses on the production and therapeutic use of phage cocktails for specific pathogenic bacteria. The Hirszfeld Institute supports the development of a more personalized phage therapeutic approach [98]. Both institutions have demonstrated convincing success rates in the use of phage therapy. Data from clinical trials at the Eliava Institute showed that phage therapy was approximately 70% and 55% effective in the treatment of *Staphylococcal* infection and *Staphylococcal* sepsis, respectively [69]. The Hirszfeld Institute pointed out that the good response rate to phage therapy was about 40% [97]. However, only a few phage therapies currently have completed phase I and phase II clinical trials, both of which lacked strict criteria and did not have evidence of properly controlled clinical trials [99]. The reports from Poland, Georgia, and other former Soviet Union countries have provided good experiences, but this form of treatment still lacks modern, properly controlled, double-blind clinical trials, and an appropriate framework is needed to make phage therapy a reality. Although phage-based clinical products have been commercialized in some eastern European countries, in western Europe and the United States, phage products for clinical applications have not yet reached the commercialization stage [7]. Currently, sporadic applications of bacteriophage therapy are carried out in western countries, where it is allowed to be used as a placebo in clinical treatment when antibiotic therapy is ineffective. Its research process and application are supervised by the Ethics Review Board and are generally protected by Article 37 of the New Helsinki Declaration [100]. The Declaration, although not a document with the force of law, became the standard for medical research ethics and serves as the basis for the development of other international guidelines. In fact, an increasing number of patients have been treated on an emergency IND (eIND) basis with the approval of the U.S. Food and Drug Administration (FDA) or the European Medicines Agency [101]. Belgium is currently implementing a pragmatic framework for phage therapy that centers on the magistral preparation (compounding pharmacy in the US) of tailor-made phage medicines [102]. This made Belgium the only western country that routinely produced phage drugs under prescription.

D’Herelle successfully treated four patients using phage injection in 1919. The earliest clinical use of phages in China can be traced back to 1958. As phage research develops in China, the cooperation between China and western countries in this field is closer. In 2022, Professor Martha Clokie from the University of Leicester, UK, in collaboration with the Institute of Animal Husbandry and Veterinary Medicine of Shandong Academy of Agricultural Sciences, published a book named *Phage Pharmacology and Experimental Methods* [103]. The book focused on pharmacology and had a strong practical application of phages. Since 2018, the Shanghai Institute of Phage and Drug Resistance (China) has conducted clinical trials on phage therapy. As far as we know, the patients receiving phage therapy must meet the following conditions:(1)The patients were between 14 and 90 years of age who suffered from multi-drug resistant bacterial infections.(2)The pathogenic bacteria that the patient is infected with should meet the following conditions:The patient is infected with pathogenic bacteria that are fully resistant to antibiotics.Pathogenic bacteria are resistant to key antibiotics (WHO has published a list of 12 superbug categories for 2017, as seen in Figure 1).The pathogen is sensitive to the key antibiotic but this antibiotic treatment is ineffective.(3)To receive at least one or more phages capable of lysing drug-resistant bacteria.(4)Be able to administer the phage by topical application, focal spraying of infection, nebulized inhalation, perfusion, local injection, or infusion tube.

## 7. The Direction of Phage Application

Due to the emergence of drug-resistant bacteria, researchers reattached importance to the study of bacteriophages as biological antibacterial agents. At present, the research and development directions of phage are as follows: strong lysis ability, good environmental adaptability and stability, no virulence genes on the genome, easy isolation and purification, no negative effect on the human immune system, and establishment of scientific pharmaceutical standards for phage preparations. Phage cocktail preparations are commonly used in clinical settings to respond to infections with different bacteria. Phage cocktails can broaden the host spectrum and reduce the frequency with which bacteria develop resistance to phages [104]. Phage cocktail therapy was approved by the U.S. government and its regulators as the first phage product for agriculture in 2005. The Shanghai Institute of Phage and Drug Resistance in China has also used phage cocktail therapy to cure patients with multi-drug resistant bacterial infections. The genetic modification of phages can broaden the host spectrum of phages, reduce cytotoxicity, and prolong the cycle time of phages in vivo. The genetic modification of phages is full of infinite possibilities to overcome some limitations of natural phage applications and improve the performance and therapeutic efficacy of phages. In addition, appropriate genetic modification strategies were used to extract lytic enzymes. Phage lysozyme is also one of the directions. The advantages of lysin lie in its easy purification, easy directional operation, high specificity, no damage to normal bacteria, and good antibacterial activity. Phages and antibiotics have different bactericidal mechanisms. The combination of the two could also improve antibacterial efficacy and produce synergistic therapeutic effects on bacteria. Genetically modified phages were used to increase or inhibit bacterial susceptibility to antibiotics. The introduction of *rpsL* and *gyrA* genes into the genome of drug-resistant *E. coli* using bacteriophage λ restored the susceptibility of the host bacteria to streptomycin [105]. Particularly, the combination of genetically modified phages and antibiotics can broaden the host lytic spectrum, which is one of the important directions for the development of phage therapy in the future. Phage has been properly investigated for its clinical potential for a long time. The upcoming years could be seen as a major milestone for phage and endolysin in terms of clinical recognition as viable alternatives to antibiotics. In order to bridge the gap and increase the packing of antibacterial drugs, researchers and clinicians must explore improved delivery strategies and creative approaches to designing phage and endolysin, as well as advancements in clinical trials [106].

## 8. Conclusions

Due to the widespread prevalence of multidrug-resistant bacteria and the slow development of antibiotics, the application of bacteriophages in the prevention, control, and treatment of bacterial infections has attracted people’s attention again. The existing research showed that phage therapy had considerable prospects and effects in clinic trials, veterinary medicine, food, and other fields. An ideal phage for clinical trials should have strong lysis ability, no toxin gene, easy separation and purification, good environmental adaptability and stability, and be harmless to the body. However, phage has some limitations, such as immune response, narrow host spectrum, dosage of administration, and preservation conditions. These problems need to be solved urgently. We can use molecular biology technology to modify the phage genome, expanding the host spectrum, improving the lytic ability and modifying the lysin. Additionally, global criteria for clinical use of phage need to be developed as soon as possible so that phages can be used more widely. It is hoped that on the basis of the existing research, we can break through the current limitations and make bacteriophages safe, green, and effective products for drug-resistant bacteria.

## Figures and Tables

**Figure 1 antibiotics-12-00417-f001:**
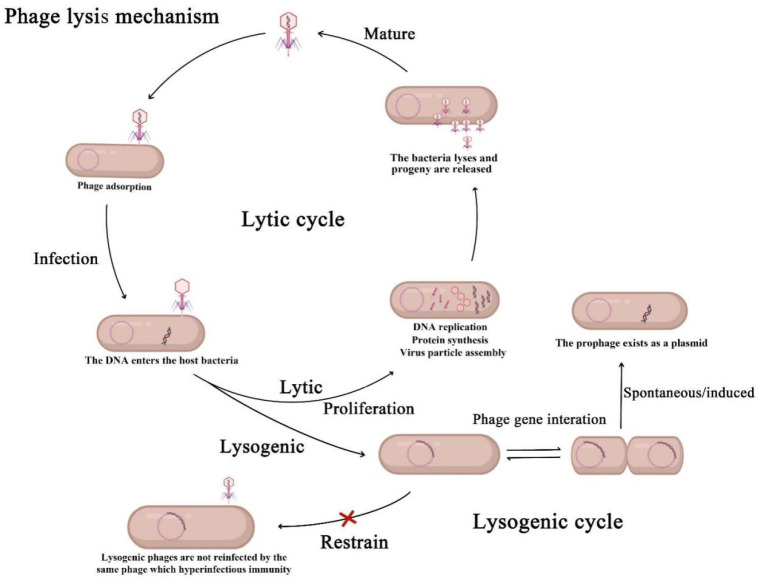
Mechanisms of action of lytic phage and lysogenic phage.

**Figure 2 antibiotics-12-00417-f002:**
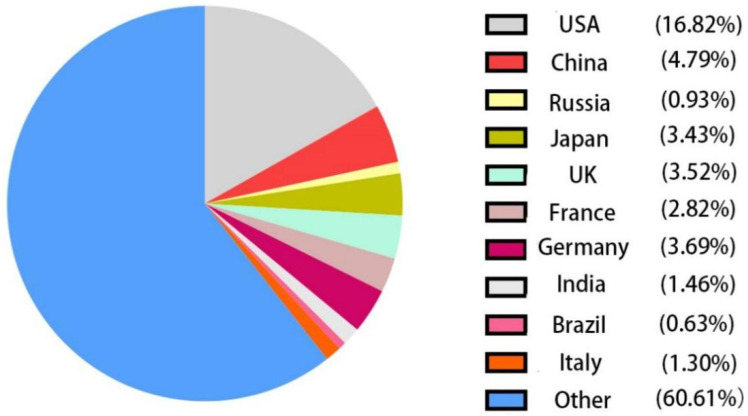
Percentage of articles published on the National Center for Biotechnology Information (NCBI) database.

**Figure 3 antibiotics-12-00417-f003:**
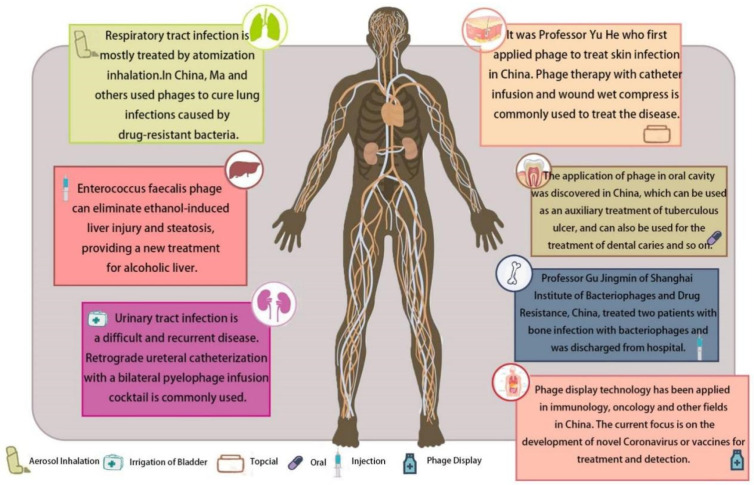
Summary of the clinical application of bacteriophage in China. Includes respiratory [72], skin [73], mouth [74], liver [75], urinary tract [76], bones, and phage display [77].

**Table 1 antibiotics-12-00417-t001:** WHO pathogen priority list, 2017.

Priority	Pathogen Name	Resistance
Critical	*Acinetobacter baumannii*	Carbapenem
	*Pseudomonas aeruginosa*	Carbapenem
	*Enterobacteriaceae*	Carbapenem and produces extended-spectrum lactamase (ESBL)
High	*Enterococcus faecalis*	Vancomycin
	*Staphylococcus aureus*	Methicillin and Vancomycin intermediary
	*Helicobacterpylori*	Clarithromycin
	*Campylobacter*	Fluoroquinolones
	*Salmonella*	Fluoroquinolones
	*Neisseria gonorrhoeae*	Cephalosporin and Fluoroquinolones
Medium	*Streptococcus pneumoniae*	Penicillin
	*Haemophilus influenzae*	Ampicillin
	*Shigella*	Fluoroquinolones

Note: Critical priority pathogens are bacteria that cause severe infection and high mortality in hospitalized patients; High priority pathogens are bacteria that cause a large number of infections in healthy people; Medium priority pathogens are three types of bacteria that are developing more resistance to available drugs.

**Table 2 antibiotics-12-00417-t002:** Summary of Bacteriophage Therapy for Bacterial Infection in Animal Models in China.

**Host Organism**													**Animal**										**Infection**												**Delivery** **b-Bacterial** **p-Phage** **bp-Bacterial and Phage**						**Phage Log (CFU)**	**Phage Log (CFU)**	**Phage Species**	**Phage Therapy Outcome** A **Remarkable Effect** **B Medium Effect** **C No Effect**					**Reference**
*Escherichia coli*	*Pseudomonas aeruginosa*	*Aerominas saimonicida*	*Klebsiella pneumoniae*	*Highly pathogenic Saimonella*	*Acinetobacter baumanii*	*Saimonellla*	*Staphylococcus aureus*	*Pasteurella*	*Streptococcus agaiactiae*	*Citrobacter freundii*	*Yersinia*	*Enterococcus faecalls*	Mouse	Bird	Chicken	Calve	Mink	Turbot	Duck	Rabbit	Tilapia mossambica	Crap	Colibacillosis	Mastitis	Pneumonia	Inflammation	Bacteremia	Pullorum disease	Sepsis	Placenta and endometritis	Salmonellosis	Systemic	Intestine	Nephritis	Take Orally	Transnasal Entry	Intraperitoneal Injection	Intramuscular Injection	Breast Injection	Intestinal Injection				Reduce Bacterial Concentration	Clear Infection	Effect of Delay in Treatment	Dose Related Studies	Continuous Injection of Phage	
◆																◆							◆												bp						10.0	11.0	4	B	B				[21]
◆																◆								◆													bp				1.8	8.7	3	A	A				[22]
		◆																◆								◆											bp				4–6	2.9–6.9	1	A	B		A		[23]
	◆												◆											◆													bp				5.3	6.3	1	B	B				[24]
◆													◆										◆												b		p				8.0	5–7	1	A	B		A		[25]
			◆										◆														◆										bp				7.3	6–7	1	A	B		A		[26]
				◆											◆													◆							bp						6.3	7.0	1	B	B				[27]
◆																			◆				◆															bp			5.3	8.5	1	B	B			A	[28]
					◆								◆																							bp					8.3	9.3	1	A	A	A			[29]
					◆								◆																◆								bp				7.7	8.0	1	A	A				[30]
	◆																◆								◆											bp					10.0	5–7	1	A	A		A		[31]
						◆									◆																		◆		bp						7.7	4.9–8.9	1	A	A		A		[32]
						◆							◆																		◆						bp				9.3	10.3	1	B	B			A	[33]
							◆													◆					◆											bp					8.5	6.5–9.5	1	B	B		A		[34]
								◆					◆																			◆					bp				1.9	8.0	1	B	B		A	A	[35]
							◆						◆											◆															bp		4.8	7.3	2	B	A				[36]
			◆										◆												◆											bp					8.3	7.3–9.3	1	B	A		A		[37]
◆													◆														◆										bp				8.0	5.0	2	A	A				[38]
◆																				◆													◆		bp						10.0	11.0	1	A	A				[39]
									◆												◆													◆						bp	5.1	9.8	1	A	B				[40]
								◆					◆												◆												bp				4.5	8.0	1	B	B				[41]
	◆												◆														◆												bp		7.5	9.8	1	A	A	A			[42]
										◆												◆					◆											bp			9.3	7.0	1	C	B		A		[43]
			◆										◆														◆											bp			8.4	3.5–4.5	3	A	A	A	A		[44]
						◆							◆																				◆					bp			8.7	9.5	1	B	B				[45]
						◆							◆																	◆								bp			3.3	5.0	1	A	A				[46]
											◆		◆																				◆			bp					9.4	9.3	1	A	A	C	A		[47]
												◆	◆														◆											bp			9.3	5.6	1	A	A		A		[48]

Note: The site where ◆ appeared is indicated as that Host Organism/Animal/Infection.

**Table 3 antibiotics-12-00417-t003:** Summary of the clinical treatment of bacterial infections by bacteriophages in China.

No.	Age–Year	Host Organism	Type of Infection	Delivery	Phage(s) Used	Outcome
[1]	66	MDR *Klebsiella pneumoniae*	Urinary tract infections	Irrigated simultaneously via the kidney and bladder	A two-phage cocktail (ΦJD902 + ΦJD905) and a three-phage cocktail (ΦJD905 + ΦJD907 + ΦJD908), combined with antibiotic treatment	Discharged and did not recur after two months of follow-up
[2]	65	Complex pan-resistant *Klebsiella pneumoniae*	Urinary tract infections	Bladder infusion	A four-phage cocktail (117, 135, 178, and GD168 phage) and a three-phage cocktail (130, 131, and 909 phage)	Pan-resistant *Klebsiella pneumoniae* was cleared and bladder infection was significantly improved.
[3]	Patient 1: 62 Patient 2: 64 Patient 3: 81 Patient 4: 78	Carbapenem-resistant *Acinetobacter baumannii* (CRAB)	Pulmonary infection	Via nebulization	A phage cocktail (2Φ)	Patient 1 and Patient 2: Discharged Patient 3: CRAB was eliminated but an un-subdued Carbapenem-resistant CRKP infection was followed and died on day 10 Patient 4: Discharged from ICU Day 7, however died of respiratory failure a month.
[4]	54	*Klebsiella pneumoniae*	Pulmonary infection	Via nebulization	Single-phage preparation (Φ59)	The symptoms of cough and expectoration were improved, and the inflammatory reaction was reduced
[5]	82	Carbapenem-resistant *Acinetobacter baumannii* (CRAB) and Carbapenem-resistant *Pseudomon asaeruginosa* (CRPA)	Pulmonary infection	Via nebulization	A two-phage cocktail (ΦPA3 + ΦPA39) and single-phage preparation (ΦAB3), combined with antibiotic treatment	The pulmonary infection was significantly improved
[6]	88	Carbapenem-resistant *Acinetobacter baumannii* (CRAB)	Pulmonary infection	Via nebulization	Single-phage preparation (Ab_SZ3), combined with antibiotic treatment	CRAB was cleared and the pulmonary infection was significantly improved

## Data Availability

No new data were created or analyzed in this study. Data sharing is not applicable to this article.

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
