# Peer review of "Bacteriophage Therapy as an Application for Bacterial Infection in China"

_antibiotics, 2023, doi:10.3390/antibiotics12020417_

Round 1
Reviewer 1 Report
Peer-review report of the review article (antibiotics-2151681)
The manuscript entitled, “Bacteriophage Therapy as An Application for Bacterial Infection in China” is a good and comprehensive review based on an excellent idea submitted for publication in the journal “antibiotics.”
This manuscript needs a major revision before acceptance.
The identification of substances having anti-pathogenic properties is a great need of the time and has a great potential to attract scientists and related groups. Keeping in view the importance of this topic, the review report of the aforementioned article is as follows.
The Idea: The idea of this review article is well-conceived and publishable. However, the facts and research are presented in a broad and generic way. The specificity of the facts is a key factor that must be considered in compiling past research in one manuscript.
Abstract: The abstract is lacking a summary of the whole manuscript, especially the key facts. The abstract is more like a summary of the introduction rather than the whole manuscript.
The method of review is not presented clearly. The inclusion or exclusion criteria are not properly given. The article needs a proper method of review to demonstrate the aim.
Introduction: This review is structured to fulfill the requirements of a review article. However, background specificity regarding the synopsis of the work is missing.
Line 23. What kind of environmental pollution is caused by antibiotics?
Line 24. What are the major health risks?
Line 26. “…more than half of the Staphylococcus…” What is the nearest possible number of such species in this genus that are resistant to antibiotics?
Line 26. What is the meaning of multiple resistances?
Lines 27-28. “…deaths caused by drug-resistant bacteria…” Which bacteria is a leading cause of such deaths?
Line 30. “…a list of global priority pathogens comprising 12 species of bacteria…” Mention names of bacteria in this list as these are of utmost importance regarding human health.
Line 39. “…large-scale animal husbandry to cut off pathogen infection…” How is large-scale animal husbandry connected to the use of bacteriophages?
Line 45. “Bacteriophage as a traditional antimicrobial therapy…” How can the bacteriophage be defined as a “traditional therapy,” and when did it come into commercial use?
Overview of phages
Lines 56-58. Different types of bacteriophages are given. How many geneses of pathogenic bacteria can they infect? Which of these types is the most efficient?
Lines 64-70. Different life cycles of bacteriophages are mentioned. Which types of bacteriophages are more virulent?
Figure 1. The quality of the image is low. The reference to this image is missing.
Phage therapy in animal models
In this section, incomplete information is given. How many articles were about phage therapy in animal models? What are the inclusion and exclusion criteria?
Line 74. “…more than 95,400 articles about phage research on NCBI…” Were these articles about phages related to animal models as correlated to the heading of this section, or do these include every article about phages?
Line 75. “…4,500 were published by Chinese…” By Chinese students, scientists, or what?
Figure 2. “…published on NCBI…” Is NCBI a publisher or a database? The caption of the image is not written properly. No proper information can be extracted from the given caption.
Table 1. The use of symbols is confusing. If the data in each row is related to one phage that is killing a microorganism in a particular organ of an animal, then why different symbols are used? These symbols could be used for different phages but not the data related to a particular phage.
Table 1. In the “infection” column, the types and sites of infection are mixed. Write either the types of infection or the sites of infection.
Table 1. In the “Delivery” column, does “bp” mean bacterial phage? If yes, write it “bp: bacterial phage,” not the other way around.
Mammals
Lines 89-90. “…mice infected with multiple drug-resistant Klebsiella pneumoniae bacteria for 2 h…” Do multiple bacteria mean various strains? If yes, which strains are included? Is 2h incubation period?
Lines 95-96. “If phage was injected 15 and 30 min after the bacteria infection…” Does direct injection not leads to the production of antibodies are phages have protein capsules? If not, why does the animal not produce antibodies?
Line 96. “…the survival rate could reach 100%...” Is the result not proven in the original experiment? The word “could” implies that there is a possibility of not reaching up to 100%.
Line 112. “…have also contributed to the application of phage and lysin therapy…” What was the contribution?
Line 116. “…adding phage in diets could improve growth performance and reduce diarrhea…” How are the phages doing that? Are the phages not supposed to be target specific, which is killing their targets?
Lines 117-119, “…the phage supplement can improve the morphological structure and function of intestines, regulate the activity of intestinal digestive enzymes, improve the structure of intestinal microorganisms, and promote the intestinal health of piglets…” Please explain the mechanisms of such claim as these are very different from the nature of a bacteriophage. How are they exerting such meliorating effects on the animals’ health?
Line 123, “…nanoparticles were effective in eliciting the antigen-specific immune responses…” What was the effect, and how did it happen?
Line 124. “…invented a bacteriophage cocktail in 2021 for clinical dairy cow…” Which phages were included in this cocktail? Were they of the same type or different types? Were they targeting the same pathogen or different pathogens?
Lines 126-128. “There were many other experiments on the successful treatment of bacterial infection in mammals by phage, but we have not given any more examples. In the future, if the immune and stress factors were excluded, the application of phage in the treatment of animal infection would have great potential.” What is the logic behind this notion? The meaning of this statement is not clear.
Ovipara
Line 131. “Inoculation of the phage suspension prior to infection…” Does it mean giving phage to the animal before infection? If yes, why is it necessary? How long can it stay effective to fight against an upcoming infection? Can it be regarded as a vaccine?
Line 134. “…infection caused by lethal E. coli.” Which strain of E. coli?
Line 135. “…that is harmful to the poultry industry…” How is it harmful?
Line 141. “…a peak reduction in S. typhimurium in ready-to-eat (RTE) duck meat.” How is it related to the present review? Can a live animal be regarded as “RTE” or this statement is about the meat after slaughter?
Aquatilia
Lines 149-151. “… Jilin Agricultural University, committed to the research of phage therapy found that bacteriophage had good therapeutic effects in the treatment of aquatic drug-resistant bacteria…” What are their findings? Their findings are missing in the paragraph.
Line 153. “…effectively counteracted the lethal dose of C. fredii…” What is the lethal dose? Line 155-157. “…phage therapy was a good method to inhibit the production of phage-resistant strains…” How can phage therapy inhibit the production of phage-resistant strains?
Line 159. “…a 12-day consecutive injection of PZL-Ah152…” What was the status of antibody production?
Line 160. “…the main organs of the treated animals.” Which organs were observed? Which animals were selected?
Lines 168-169. “…phage qdvp001 could purify V. parahaemolyticus in oysters cultured environment and also in oyster body…” What does the word “purify” refer to? Is the phage killing the target or separating it from the environment?
Lines 171-172. “…in a variety of marine vertebrates and invertebrates…” Which ones?
Lines 175. “…the phages could enhance the resistance of turbot…” How can it enhance resistance?
Application of bacteriophages in clinic
This section explains the use of phage treatment in human subjects. However, the dose of application is seldom mentioned, which is crucial to understand the working of the bacteriophages. It is suggested to include the doses of phage application throughout the manuscript.
Line 186. “…treat clinical diseases also has a long history in China.” How long? Which diseases were treated? How frequently was the phage treatment used?
Line 187. “…the first to develop phage research and related production in China.” When did they develop the first product? Whom was it efficient against?
Lines 187-189. “The Wuhan Institute of Biological Products also carried out a period of trial production around 1958. In 2017, Shanghai Institute of Phage was established…” What is the status of research between 1958 and 2017?
Line 195. “…non-sensitive antibiotic-phage combination…” How does it work? Why is it efficient?
Respiratory tract infection
Line 224. “…using phage to treat drug-resistant bacteria infections…” Which strains of bacteria were treated?
Lines 226-227. “…patients hospitalized with critical COVID-19. They suggested the potential of phages on rapid responses to secondary CRAB outbreak in COVID-19 patients.” What is the relationship between phage therapy and COVID-19?
Line 228. “…a specific lysing pathogen-specific phage…” Which pathogen was targeted? Urinary tract infection
Line 240-241. “The combination of antibiotic with a phage mixture inhibited the appearance of phage-resistant mutants…” Which antibiotics were included?
Others
Lines 252-253. “…significantly shortened the clinical treatment period…” How much shortening was observed?
Lines 253-254. “…using E. faecalis phage can significantly reduce liver cytolytic level and fecal enterococcus quantity…” What mechanism did the phage use?
Limitation of phage therapy
In this section, the authors have explained the limitation of phage therapy in detail. It is suggested to include a few words about the effective period of the phages, whether can stay active for a long period or not. Moreover, how can a phage work in deeper layers of the body? Viruses have a great potential to mutate. How can his problem be addressed in the case of phages?
Regulation of phage application in China and Western countries
In this section, the regulations are given. The reviewer is unable to find any information regarding China in this section. As the review article is focused on phage research and application in China, Chinese regulations must be included.
Facts and data inclusion
Throughout the manuscript, the efficiency of the phages is elaborated by words including, “significant, increase or decrease, efficient,” and others. However, the exact values are rarely proven by data and figures. Several examples are mentioned above. Every statement and mechanism should be confirmed by a specific figure, name, or mechanism. Recheck the whole manuscript, and add required figures, names, or mechanisms to make the claims believable.
The repetition of facts is present. If a few studies show similar results, it is better to merge them into one sentence. The whole manuscript must be concise and well-presented.
Formatting
The manuscript must be thoroughly formatted as several inconsistencies are found throughout the manuscript.
A similar terminology or name must be used according to the author's guidelines. Several inconsistencies are found regarding species names. A consistent name for the same substance must be used.
The names of the bacterial species are inconsistently formatted. Somewhere genus names are shortened, whereas these are left as a full name.
Language: The language and grammar of the manuscript must be improved. The use of words must also be improved. In case of multiple figures regarding different facts are given, the word “respectively” should be used.
Author Response
To Editor and Reviewers:
Thank you for giving us a chance to revise the manuscript. We also thank the reviewers for constructive suggestions to help us improve the quality of the manuscript. We have made changes in the manuscript according to the reviewers’ comments and corresponding to your advice. We hope these changes make the manuscript acceptable for publication.
Response to Reviewer 1 Comments
Point 1: Abstract: The abstract is lacking a summary of the whole manuscript, especially the key facts. The abstract is more like a summary of the introduction rather than the whole manuscript.
Response:
The method of review is not presented clearly. The inclusion or exclusion criteria are not properly given. The article needs a proper method of review to demonstrate the aim.
Introduction: This review is structured to fulfill the requirements of a review article. However, background specificity regarding the synopsis of the work is missing.
Response 1:Agreed. We are really sorry to bring you the review trouble. According to your suggestion, we have rewritten this abstract and make summary of the whole manuscript, as seen from the lasted revised manuscript. We found 95000 articles related to bacteriophages through keyword search in the National Center for Biotechnology Information (NCBI) database (https://www.ncbi.nlm.nih.gov/), of which more than 4500 were from Chinese studies. Some of the 4,500 articles are related to the study of phage biology, including genome sequencing analysis. Not all articles are relevant to the use of phage therapy in animal models and clinical treatment. At the same time, we also refer to the articles on China National Knowledge Infrastructure (CNKI, https://cnki.net/) accordingly. We only cited about 45 articles on animal models and human clinical, which can well explain the progress of Chinese scientific research in mammals, egg-laying animals, aquatic animals and human clinical, so we selected these articles and cited them in our review.
Point 2: Line 23. What kind of environmental pollution is caused by antibiotics?
Response: Thank you for your comments. Due to abuse and insufficient treatment, a large number of antibiotics enter the water environment every year, resulting in the phenomenon of “false persistence”. SAs are generally detected in the groundwater in most parts of the United States, and the peak concentrations of sulfamerazine (SM1) and sulfamethazine (SMZ) are as high as 360 ng/L and 1100 nm/L [1], respectively. In addition, antibiotics such as SAs, MLs, and FQs were detected in groundwater near livestock farms [2]. The peak concentrations of SAs and MLs in groundwater in Barcelona, Spain, reached 37.1 ng/L and 2980 ng/L [3], respectively. SAs were generally detected in groundwater near a large-scale farm in Germany [4]. Antibiotics in the water environment will lead to the imbalance of microbial populations, and even induce the generation of drug resistance genes, and, at the same time, they will enter the human body with the accumulation in the food chain, threatening human health [5-8].
Point 3: Line 24. What are the major health risks?
Response:Thanks for your careful review. Antibiotics have now been the main treatment for bacterial diseases. However, when bacteria become resistant to antibiotics, commonly used antibiotics become ineffective. This can lead to fatal diseases such as fatal pneumonia, pyaemia, septicemia, etc. For example, Klebsiella pneumoniae, is one of the important pathogens which can lead to septic shock, multi-organ failure, and even death. Bloodstream infection with carbapenem-resistant Klebsiella pneumoniae has a case fatality rate of up to 40% within 30 days [9].
Point 4: Line 26. “…more than half of the Staphylococcus…” What is the nearest possible number of such species in this genus that are resistant to antibiotics?
Response: Thank you for your comment. This data we found from this reference [10]. However, we did not find out what the specific data is by reviewing the literature. Since bacteria are prone to antibiotic resistance, we speculated that it may be Staphylococcus aureus or Streptococcus, but no specific data is given in this article.
Point 5: Line 26. What is the meaning of multiple resistances?
Response:Thank you for your comment. Staphylococcus aureus is a superbug due to its susceptibility to resistance to many antibiotics. According to a survey, among the 1326 strains of Staphylococcus aureus detected from clinical in 2019 to 2021, more than 95% were resistant to penicillin, more than 50% were resistant to erythromycin and clindamycin, and the resistance rate of tetracycline was also about 20%, and resistance to antibiotics such as gentamicin and ciprofloxacin was also found [11]. At present, most bacteria are insensitive to more than three antibiotics. And we found through our review that “multiple” may not be working properly, so we changed it to “multidrug”. As seen from line 33, page 1 in revised manuscript (with revisions tracked).
Point 6: Lines 27-28. “…deaths caused by drug-resistant bacteria…” Which bacteria is a leading cause of such deaths?
Response: Thank you for your comment We got this fact by consulting the literature [12], and the author did not give a specific analysis of the data. However, we found out by looking up the literature, such as Klebsiella pneumoniae, is one of the important pathogens that cause bloodstream infections. Klebsiella pneumoniae bloodstream infection (KPBSI) progresses rapidly and in severe cases can lead to septic shock, multi-organ failure, and even death. Studies have shown that the 30-day case fatality rate of patients with bloodstream infection Carbapenem-resistant klebsiella pneumoniae is as high as 40%, which poses a great threat to patient survival and poses a great challenge to clinicians [9].
Point 7: Line 30. “…a list of global priority pathogens comprising 12 species of bacteria…” Mention names of bacteria in this list as these are of utmost importance regarding human health.
Response:Agreed. Special thanks for your careful review. According to your suggestion, we have added a table. As seen from line 73-77, page 2 in revised manuscript (with revisions tracked).
Point 8: Line 39. “…large-scale animal husbandry to cut off pathogen infection…” How is large-scale animal husbandry connected to the use of bacteriophages?
Response:Thank you for your comments. At present, bacteriophages have many connections with large-scale animal husbandry. For example, they can be prepared as bacteriophage spray preparations to disinfect and sterilize the environment of large-scale farms, which can effectively prevent animal diseases, reduce the presence of pathogens in the environment, and reduce the viral load in animals.[13] And the addition of bacteriophages to piglet feed has a positive effect on intestinal inflammation, intestinal barrier function and intestinal flora in weaned piglets [14].
Point 9: Line 45. “Bacteriophage as a traditional antimicrobial therapy…” How can the bacteriophage be defined as a “traditional therapy,” and when did it come into commercial use?
Response: Thank you for your comment. In 1917, Felix d'Herelle was the first to extract the concept of the word "bacteriophage" from the Latin etymology, which Chinese interpreted as "swallowing bacteria." Félix d'Herelle began treating dysentery caused by Shigella with bacteriophages in 1919. Bruynoghe et al. treated skin infections caused by Staphylococci with bacteriophage preparations in 1921. However, since 1940, with the rise of the antibiotic industry, especially the popularity of sulfonamides, phage preparations popular in Europe have been crowded out, and later tended to be withdrawn from the market. But the Eliava Institute in Georgia has never given up on the use of phage therapy and has successfully cured many cases of urinary tract infections, cystitis and prostatitis.
The commercial application of bacteriophages has not been widely promoted around the world, because it is difficult to apply for intellectual property protection. However, through our many searches, The Soviet Union and some countries in Eastern Europe have been using phage therapy on a large scale, and there were 5 research institutes in the Soviet Union engaged in the production of phage preparations. The world's major pharmaceutical companies (such as Eli lilly) also produced and sold phage preparations before 1940.
Point 10: Lines 56-58. Different types of bacteriophages are given. How many geneses of pathogenic bacteria can they infect? Which of these types is the most efficient?
Response: Thank you for your comment. We are very sorry that the data on your question is difficult to count. At present, the order of magnitude of bacteriophages has reached 1013, and each bacteriophage can specifically infect one or more pathogenic bacteria. Due to the special structure of the phage tail filament protein, it can bind specifically to bacterial surface receptors. So, we don't know exactly which genes are associated with pathogenic infections.
Which type of bacteriophage is more effective is impossible to measure. According to the different results of the effect of bacteriophages on bacteria, it is divided into virulent phages and lysogenic phages [15]. The virulent phage invades the host bacteria and uses the raw materials provided by the host cells for replication and reproduction, and then lyses the host. Lysogenic phage integrates nucleic acids into the host's genome and replicate with the replication of bacteria. Each bacteriophage can specifically lyse one or more bacteria. But as for whether it has an effect, we need to prove it through a lot of experimentation.
Point 11: Lines 64-70. Different life cycles of bacteriophages are mentioned. Which types of bacteriophages are more virulent?
Response:Thank you for your comments. Lytic phages, also known as virulent phages, can inject their genomes into the bacteria, hijack the metabolic function of the host and lyse the host cells to produce new progeny phages. The other type of phages, lysogenic phages, have a different lifestyle and infect their host by initiating a lysogenic cycle, where the phage genome remains dormant as a prophage, replicates along with its host and occasionally bursts into a lytic cycle under a specific trigger, and exerts a lytic effect. So, there is no way to evaluate which bacteriophage is more virulent. However, we have found through a large number of literature that the phage mostly used in treatment was virulent phage.
Point 12: Figure 1. The quality of the image is low. The reference to this image is missing.
Response:Agreed. We are really sorry for the oversights in the manuscript. According to your suggestion, we have added the reference (Zhang M et al. 2020). As seen from line 92, page 3 in revised manuscript (with revisions tracked).
Phage therapy in animal models
Point 13: In this section, incomplete information is given. How many articles were about phage therapy in animal models? What are the inclusion and exclusion criteria?
Response:Thanks for your careful review. We found 95000 articles related to bacteriophages through keyword search in the National Center for Biotechnology Information (NCBI) database (https://www.ncbi.nlm.nih.gov/), of which more than 4500 were from Chinese studies. Some of the 4,500 articles are related to the study of phage biology, including genome sequencing analysis. Not all articles are relevant to the use of phage therapy in animal models. At the same time, we also refer to the articles on China National Knowledge Infrastructure (CNKI, https://cnki.net/) accordingly. We only cited about 45 articles on animal models and human clinical, which can well explain the progress of China's scientific research in mammals, egg-laying animals, aquatic animals and human clinical, so we selected these articles and cited them in our review.
Point 14: Line 74. “…more than 95,400 articles about phage research on NCBI…” Were these articles about phages related to animal models as correlated to the heading of this section, or do these include every article about phages?
Response:Thanks for your careful review. The more than 95000 articles mentioned here are articles related to bacteriophages that we found through keyword search in the National Center for Biotechnology Information (NCBI) database. Some of the 4,500 articles are related to the study of phage biology including genome sequencing analysis, not all of them contain animal experiments.
Point 15: Line 75. “…4,500 were published by Chinese…” By Chinese students, scientists, or what?
Response:Thanks for your comment. More than 4,500 articles about phage research on National Center for Biotechnology Information (NCBI) were published by Chinese researchers.
Point 16: Figure 2. “…published on NCBI…” Is NCBI a publisher or a database? The caption of the image is not written properly. No proper information can be extracted from the given caption.
Response:We are really sorry to bring you the review trouble. Data collected by using the function of the database on National Center for Biotechnology Information (NCBI, https://www.ncbi.nlm.nih.gov/). NCBI is the National Center for Biotechnology Information, in addition to building the GenBank nucleic acid sequence database (The data resources of this database are from several major DNA databases around the world, including the Japanese DNA database DDBJ, the European molecular biology laboratory database EMBL, and several other well-known scientific research institutions), NCBI can also provide many powerful data retrieval and analysis tools. NCBI has Medline's biomedical research paper indexing feature, making all of these databases accessible online through the Entrez search engine. NCBI is a worldwide representative database with a vast and thorough number of papers, we picked it as our manuscript's data source.
Point 17: Table 1. The use of symbols is confusing. If the data in each row is related to one phage that is killing a microorganism in a particular organ of an animal, then why different symbols are used?
These symbols could be used for different phages but not the data related to a particular phage.
Response:Agreed. Special thanks for your suggestion. According to your suggestion, we have revised this table. As seen from line 114-115, page 5-6 in revised manuscript (with revisions tracked).
Point 18: Table 1. In the “infection” column, the types and sites of infection are mixed. Write either the types of infection or the sites of infection.
Response:Agreed. Thanks for your constructive suggestion. According to your suggestion, we have changed to infection type. As seen from line 114-115, page 5-6 in revised manuscript (with revisions tracked).
Point 19: Table 1. In the “Delivery” column, does “bp” mean bacterial phage? If yes, write it “bp: bacterial phage,” not the other way around.
Response:Agreed. We are really sorry to bring you the review trouble. According to your suggestion, we have rewritten the “Delivery” column.
Mammals
Point 20: Lines 89-90. “…mice infected with multiple drug-resistant Klebsiella pneumoniae bacteria for 2 h…” Do multiple bacteria mean various strains? If yes, which strains are included? Is 2h incubation period?
Response:Thank you for your comment. In this reference, he used only one strain of bacteria, Klebsiella pneumoniae, which has resistance to multidrug antibiotics [16]. And it had a short latent period of 30 min and intranasal administration of a single dose of 2 × 109 PFU/mouse 2 h after KP 1513 inoculation was enough to protect mice against lethal pneumonia.
Point 21: Lines 95-96. “If phage was injected 15 and 30 min after the bacteria infection…” Does direct injection not leads to the production of antibodies are phages have protein capsules? If not, why does the animal not produce antibodies?
Response:Thank you for the comment. Injected 15 and 30 min after the bacteria infection is to give bacteria a time to augment in the body. Only when the bacteria reach a certain amount, the bacteriophage will specifically absorb the bacteria and lyse them. It takes time for the animal body to produce antibodies, generally ranging from a few days to ten days. Due to the short treatment time, there is a high probability that no antibodies will be produced. As for when antibodies are produced, this needs to be determined by test, and the relevant test is not carried out in this article.
Point 22: Line 96. “…the survival rate could reach 100%...” Is the result not proven in the original experiment? The word “could” implies that there is a possibility of not reaching up to 100%.
Response:Agreed. We are really sorry for the oversights in the manuscript. According to your suggestion, we have replaced “could” to “can”. As seen from line 126, page 7 in revised manuscript (with revisions tracked).
Point 23: Line 112. “…have also contributed to the application of phage and lysin therapy…” What was the contribution?
Response:Special thanks for your careful review. In the present study, they report a new lysin LysP53 from Acinetobacter baumannii phage 53. Their findings collectively establish that LysP53 could be a promising candidate in the treatment of topical infections caused by multiple Gram-negative pathogens [17].
Point 24: Line 116. “…adding phage in diets could improve growth performance and reduce diarrhea…” How are the phages doing that? Are the phages not supposed to be target specific, which is killing their targets?
Response: Thank you for your comment. After weaning, the piglets experience a decrease in digestion, resulting in diarrhea. About this, the author discusses in his article [14]: Compared with the control diet, dietary 400 mg/kg bacteriophage inclusion increased average daily gain and average daily feed intake, and decreased feed/gain ratio and diarrhea incidence of weaned piglets (P < 0.05). MiSeq sequencing analysis showed that bacteriophage regulated the microbial composition in caecum digesta, as indicated by higher observed_species, Chao1, and ACE richness indices, as well as changes in the relative abundance of Firmicutes, Bacteroidetes, and Tenericutes (P < 0.05). In this article by the author, this phenomenon is not specifically explained. We also can't give specific inferences about why the phenomenon of the growth improvement and diarrhea reduction occurred.
Point 25: Lines 117-119, “…the phage supplement can improve the morphological structure and function of intestines, regulate the activity of intestinal digestive enzymes, improve the structure of intestinal microorganisms, and promote the intestinal health of piglets…” Please explain the mechanisms of such claim as these are very different from the nature of a bacteriophage. How are they exerting such meliorating effects on the animals’ health?
Response:Thanks for your comment. Because this experiment was not done by us, the changes of bacteriophages and intestinal bacteria are changeable, and we can't explain it to you very clearly. This phenomenon is not unique to this bacteriophage, and some articles have confirmed that the addition of bacteriophages can improve the intestinal. By consulting the literature, we can provide you with the following points, and then let you better review the manuscript. Recent sequencing-based approaches show that gut bacteriophages profoundly influence gut physiology [7, 8] and host health [18,19]. The relationship between gut bacteriophages and bacteria is dynamic. During early intestinal development, it consists mostly of predator and prey interactions characterized by rapid fluctuations in bacterial and phage populations in terms of both abundance and diversity [20-22]. Since the resident bacterial community is unstable in the early stages of intestinal development, bacteriophage infection at this stage rapidly decreases the prey population, which allows competitive bacteria to colonize the gut, and then leads to the increase of new bacteriophages with a new infection cycle [23]. Furthermore, bacteriophages provide defense against bacterial pathogens through lytic infection [24] and can adhere to intestinal mucosal surfaces, where they are more likely to encounter and kill invading bacteria [25]. Therefore, bacteriophages play a pivotal role in establishing the early gut microbiota.
Point 26: Line 123, “…nanoparticles were effective in eliciting the antigen-specific immune responses…” What was the effect, and how did it happen?
Response:Thank you for your comment. Through our literature review, the author's interpretation of this is “we demonstrated the potential of T7 bacteriophage-based nanoparticles displaying a genetically fused G-H loop peptide (T7-GH) as a FMDV vaccine candidate. Recombinant T7-GH phage was constructed by inserting the G-H loop coding region into the T7 Select 415-1b vector. Purified T7-GH phage nanoparticles were analyzed by SDS-PAGE, Western blot and Dot-ELISA. Pigs seronegative for FMDV exposure were immunized with T7-GH nanoparticles along with the adjuvant Montanide ISA206, and two commercially available FMDV vaccines (InactVac and PepVac). Humoral and cellular immune responses, as well as protection against virulent homologous virus challenge were assessed following single dose immunization. Pigs immunized T7-GH developed comparable anti-VP1 antibody titers to PepVac, although lower LPBE titers than was induced by InactVac. Antigen specific lymphocyte proliferation was detected in T7-GH group similar to that of PepVac group, however, weaker than InactVac group. Pigs immunized with T7-GH developed a neutralizing antibody response stronger than PepVac, but weaker than InactVac. Furthermore, 80% (4/5) of T7-GH immunized pigs were protected from challenge with virulent homologous virus. These findings demonstrate that the T7-GH phage nanoparticles were effective in eliciting antigen specific immune responses in pigs, highlighting the value of such an approach in the research and development of FMDV vaccines" [20].
Point 27: Line 124. “…invented a bacteriophage cocktail in 2021 for clinical dairy cow…” Which phages were included in this cocktail? Were they of the same type or different types? Were they targeting the same pathogen or different pathogens?
Response:Special thanks for your good comments. The bacteriophage cocktail preparation for the treatment of cow mastitis comprises the bacteriophages Ecp1, Ecp3, Ecp5, Ecp16, Ecp17 and vB_ EcoM_ XJ2 of Escherichia coli from cow mastitis, bovine mastitis streptococcus bacteriophage vB_ StrM_ L1、vB_ SagS_ FSN1, Staphylococcus aureus phage P42, vB from cow mastitis_ Sau S_ IMEP5. And we have briefly described this in the revised manuscript. As seen from line 154-156, page 7 in revised manuscript (with revisions tracked).
Point 28: Lines 126-128. “There were many other experiments on the successful treatment of bacterial infection in mammals by phage, but we have not given any more examples. In the future, if the immune and stress factors were excluded, the application of phage in the treatment of animal infection would have great potential.” What is the logic behind this notion? The meaning of this statement is not clear.
Response:Agreed. We are really sorry for the oversights in the manuscript. According to your suggestion, we have rewritten this sentence. As seen from line 158-159, page 7 in revised manuscript (with revisions tracked).
Ovipara
Point 29: Line 131. “Inoculation of the phage suspension prior to infection…” Does it mean giving phage to the animal before infection? If yes, why is it necessary? How long can it stay effective to fight against an upcoming infection? Can it be regarded as a vaccine?
Response:Yes, giving phage to the animal before infection. In this article, they found that the injection of bacteriophage 6 hours before infection can prevent the corresponding bacterial infection, but the researchers did not explore the duration of resistance, because the main purpose of this study is to determine that the injection of bacteriophage before infection can prevent the animal body. Because we are not specialized to study vaccines, our responses are not necessarily accurate and are for your information only to the best of our knowledge to confirm a vaccine that should detect antibody production in vivo, but antibody assays were not performed by this experimenter.
Point 30: Line 134. “…infection caused by lethal E. coli.” Which strain of E. coli?
Response:We are sorry for bring you the review trouble. The E. coli is E. coli serotype O2. And according to your suggestion, we have replaced “E. coli.” to “E. coli serotype O2”. As seen from line 163, page 7 in revised manuscript (with revisions tracked).
Point 31: Line 135. “…that is harmful to the poultry industry…” How is it harmful?
Response:Thanks for your comment. Salmonellosis has long harmed China's poultry industry. Salmonella albicans infect eggs by vertical transmission. Salmonella pullorum infection can easily cause a large number of chicks to die, the mortality rate is about 40%, which brings huge losses to ordinary production.
Point 32: Line 141. “…a peak reduction in S. typhimurium in ready-to-eat (RTE) duck meat.” How is it related to the present review? Can a live animal be regarded as “RTE” or this statement is about the meat after slaughter?
Response:Thank you for your comment. We are sorry for bring you the review trouble. We have deleted this example.
Aquatilia
Point 33: Lines 149-151. “… Jilin Agricultural University, committed to the research of phage therapy found that bacteriophage had good therapeutic effects in the treatment of aquatic drug-resistant
bacteria…” What are their findings? Their findings are missing in the paragraph.
Response:Thank you for your comment. In our manuscript, we have a few examples from their team, such as: reference [26], [27] and [28]. As seen from line 181-195, page 8 in revised manuscript (with revisions tracked).
Point 34: Line 153. “…effectively counteracted the lethal dose of C. fredii…” What is the lethal dose?
Response :Thank you for your comment. We are sorry for bring you the review trouble. The lethal dose is 1 × 109 CFU/carp. And we have added this in our manuscript, as seen from line 184, page 8 in revised manuscript (with revisions tracked).
Point 35: Line 155-157. “…phage therapy was a good method to inhibit the production of
phage-resistant strains…” How can phage therapy inhibit the production of phage-resistant strains?
Response: Special thanks for your comments. As described in the Li’s article, phage cocktails broaden the bactericidal spectrum and reduce the emergence of bacterial resistance compared to previously reported single phage preparations [29]. Similarly, Yu[27] also found through experiments that phage cocktails can inhibit the emergence of drug-resistant strains and reduce their mutation rates.
Point 36: Line 159. “…a 12-day consecutive injection of PZL-Ah152…” What was the status of antibody production?
Response:Thank you for your comments. The author does not address the level of antibody production in the article. Unfortunately, we have no way of knowing that the 12-day injection will produce antibodies, or the level of antibodies produced.
Point 37: Line 160. “…the main organs of the treated animals.” Which organs were observed? Which animals were selected?
Response: Thank you for your comment. The organs observed include liver, spleen, kidney, gut and gill. And the animal selected by author is fish.
Point 38: Lines 168-169. “…phage qdvp001 could purify V. parahaemolyticus in oysters cultured environment and also in oyster body…” What does the word “purify” refer to? Is the phage killing the target or separating it from the environment?
Response: Thanks for your comment. The author in his article used "purify". We inferred from the article to kill the pathogenic bacteria (Vibrio parahaemolyticus) in the oyster's living environment and its body.
Point 39: Lines 171-172. “…in a variety of marine vertebrates and invertebrates…” Which ones?
Response: Thanks for your careful review. In our review, carp and turbot we cited belong to vertebrates. Oysters, on the other hand, are invertebrates.
Point 40: Lines 175. “…the phages could enhance the resistance of turbot…” How can it enhance resistance?
Response: Special thanks for your careful review. According to Cui’s research [30], the phage titer in seawater was 7.87 ×106 PFU/mL, which was significantly higher than that in intestine (5.93 × 105 PFU/mL), liver (1.0 × 103 PFU/mL), kidney (2.1 × 102 PFU/mL) and blood (2.43 × 103 PFU/mL) (P < 0.05) before challenge. Seven days after V. harveyi challenge, the phage titer in the intestines of turbot increased significantly (P < 0.05), even higher than that in feed. So, the phages can enhance the resistance of turbot to V. harveyi VH5 infection.
Point 41: This section explains the use of phage treatment in human subjects. However, the dose of application is seldom mentioned, which is crucial to understand the working of the bacteriophages. It is suggested to include the doses of phage application throughout the manuscript.
Response: Thanks for your carefully review. We have added the dose of phage application in our manuscript. As seen in revised manuscript (with revisions tracked).
Point 42: Line 186. “…treat clinical diseases also has a long history in China.” How long? Which diseases were treated? How frequently was the phage treatment used?
Response: Thank you for your comments. The former Dalian Institute of Biological Products was the first to develop phage research and related production in China, which is used for the prevention and treatment of dysentery. The Wuhan Institute of Biological Products also carried out a period of trial production around 1958. Si Wandong adopted oral phages in 1995 to treat dysentery patients. In 1956, He Mingda conducted research on the outbreak of epidemic dysentery among the infantry stationed in a regiment, and tried the Shiga and Fuchs's polyvalent bacteriophages (produced by Dalian Institute in 1954) for treatment [31]. Today, some research institutes in Qingdao, Shenzhen, and Shanghai are successively carrying out phage-related research work. Therefore, we said phages have a long history in China.
Point 43: Line 187. “…the first to develop phage research and related production in China.” When did they develop the first product? Whom was it efficient against?
Response: Thank you for reading this manuscript carefully. Up to now, from the information we have found, the Dalian Institute of Biological Products has produced a batch of phage preparations for the prevention and treatment of dysentery (Shigella and F. fowleri) in 1954 [32].
Point 44: Lines 187-189. “The Wuhan Institute of Biological Products also carried out a period of trial production around 1958. In 2017, Shanghai Institute of Phage was established…” What is the status of research between 1958 and 2017?
Response: Thank you for your comments. Our search in NCBI's database by keywords found that in 1958-2017, Chinese scientists published more than 2700 phage related articles, and the accumulation of these data laid a certain experimental foundation for phage in clinical research including veterinary clinic.
Point 45: Line 195. “…non-sensitive antibiotic-phage combination…” How does it work? Why is it efficient?
Response: Thank the reviewer for the question. The author of this article explains this as follows: “After several rounds of therapeutic exploration, we found that the combination of non-sensitive antibiotics and phage cocktails produced a clear synergistic effect. And this phenomenon has been proved by subsequent experiments. At the same time, phage cocktails can limit the emergence of phage-resistant mutants and greatly improve the effect of combined drugs” [33].
Point 46: Line 224. “…using phage to treat drug-resistant bacteria infections…” Which strains of bacteria were treated?
Response: Thanks for the reviewer’s valuable comment. Through our review of this article, they treated Acinetobacter baumannii.
Point 47: Lines 226-227. “…patients hospitalized with critical COVID-19. They suggested the potential of phages on rapid responses to secondary CRAB outbreak in COVID-19 patients.” What is the relationship between phage therapy and COVID-19?
Response: Thank you for your comment. Critically ill COVID-19 patients with prolonged hospital stays were at increased risk of Secondary bacterial infections. At the beginning of March 2020, Wu et al. applied phage therapy to successfully control an outbreak of secondary carbapenem-resistant Acinetobacter baumannii (CRAB) infections in an ICU dedicated to COVID-19 patients in Shanghai, China. Both phage susceptibility testing and multilocus-sequence typing revealed identical profiles of CRAB strains present in these patients. Treatment with a 2-phage cocktail was associated with reduced CRAB burdens in all cases. These results indicated the potential of phages for rapid management of SBI outbreak in COVID-19 patients. [34].
Point 48: Line 228. “…a specific lysing pathogen-specific phage…” Which pathogen was targeted?
Response: Thank you for your comment. This bacteriophage is targeted against carbapenem-resistant Acinetobacter baumannii (CRAB).
Point 49: Line 240-241. “The combination of antibiotic with a phage mixture inhibited the appearance of phage-resistant mutants…” Which antibiotics were included?
Response: Thanks for your constructive suggestion. We apologize for not mentioning this in our manuscript. The antibiotic is Trimethoprim-sulfamethoxazole. And we added it to our manuscript. As seen from line 276, page 12 in revised manuscript (with revisions tracked).
Point 50: Lines 252-253. “…significantly shortened the clinical treatment period…” How much shortening was observed?
Response: Thank you for your comments. Our sentence is misdescribed and we have made changes. As seen from line 289, page 12 in revised manuscript (with revisions tracked).
Point 51: Lines 253-254. “…using E. faecalis phage can significantly reduce liver cytolytic level and fecal enterococcus quantity…” What mechanism did the phage use?
Response: Thank you for your comment. In the original paper, the author briefly described its mechanism [35]. Several studies have shown that gut microbes play an important role in the occurrence and development of alcoholic hepatitis, and phages, as part of intestinal microorganisms, can directly target their sensitive bacteria, while leading to cascading effects on other strains and intestinal metabolomics. Disease severity and short-term mortality in patients with AH are closely related to the presence of cytolysin-producing Enterococcus faecalis. Cytolytic does not affect the barrier function of the intestine, but increases the translocation of cytolytic Enterococcus faecalis to the liver, so phages target cytolysin-positive Enterococcus faecalis not only reduces the number of Enterococcus faecalis but also reduces the level of hepatic cytolytic.
Point 52: Limitation of phage therapy
In this section, the authors have explained the limitation of phage therapy in detail. It is suggested to include a few words about the effective period of the phages, whether can stay active for a long period or not. Moreover, how can a phage work in deeper layers of the body? Viruses have a great potential to mutate. How can his problem be addressed in the case of phages?
Response: Thank you for your positive and constructive comments on our submission. About the effective period of the phages, the manuscript has been added to Page 13, line 311 to 317 to account for this point.
Point 53: Regulation of phage application in China and Western countries
In this section, the regulations are given. The reviewer is unable to find any information regarding China in this section. As the review article is focused on phage research and application in China, Chinese regulations must be included.
Response: Thank you for your comments. China's phage research is still in the preclinical stage. Shanghai and Qingdao have established companies related to the research of bacteriophages to develop phage biological products. At present, China does not have relevant regulations. It is believed that with the gradual advancement of phage clinical trials and the continuous improvement of more pharmacokinetic data, relevant laws and regulations will be further proposed.
Point 54: Facts and data inclusion
Throughout the manuscript, the efficiency of the phages is elaborated by words including, “significant, increase or decrease, efficient,” and others. However, the exact values are rarely proven by data and figures. Several examples are mentioned above. Every statement and mechanism should be confirmed by a specific figure, name, or mechanism. Recheck the whole manuscript, and add required figures, names, or mechanisms to make the claims believable.
The repetition of facts is present. If a few studies show similar results, it is better to merge them into one sentence. The whole manuscript must be concise and well-presented.
Response: Thank you for your critical comments on this point. We have rechecked the whole manuscript, and added relevant data. As seen from the new revised manuscript (with revisions tracked).
Point 55: Formatting
The manuscript must be thoroughly formatted as several inconsistencies are found throughout the manuscript.
A similar terminology or name must be used according to the author's guidelines. Several inconsistencies are found regarding species names. A consistent name for the same substance must be used.
The names of the bacterial species are inconsistently formatted. Somewhere genus names are shortened, whereas these are left as a full name.
Response: Thank you for appreciating our efforts to address your constructive comments, which really helped us improve the manuscript. We modified the format of the relevant names in the manuscript and presented them in a highlighted form. As seen from the new revised manuscript (with revisions tracked).
Point 56: Language:
The language and grammar of the manuscript must be improved. The use of words must also be improved. In case of multiple figures regarding different facts are given, the word “respectively” should be used.
Response: Thanks for your careful review. We are really sorry to bring you the review trouble. The manuscript has been revised again by Nada Alkhorayef, Samia S. Alkhalil and Essam Eldin Abdelhady Salama who is the native English speaker. The English grammar through the manuscript has been modified. As seen from the new revised manuscript (with revisions tracked).
- Lapworth, D.J.; Baran, N.; Stuart, M.E.; Ward, R.S. Emerging organic contaminants in groundwater: A review of sources, fate and occurrence. Environ Pollut 2012, 163, 287-303, doi:10.1016/j.envpol.2011.12.034.
- Bartelt-Hunt, S.; Snow, D.D.; Damon-Powell, T.; Miesbach, D. Occurrence of steroid hormones and antibiotics in shallow groundwater impacted by livestock waste control facilities. J Contam Hydrol 2011, 123, 94-103, doi:10.1016/j.jconhyd.2010.12.010.
- López-Serna, R.; Jurado, A.; Vázquez-Suñé, E.; Carrera, J.; Petrović, M.; Barceló, D. Occurrence of 95 pharmaceuticals and transformation products in urban groundwaters underlying the metropolis of Barcelona, Spain. Environ Pollut 2013, 174, 305-315, doi:10.1016/j.envpol.2012.11.022.
- Sacher, F.; Lange, F.T.; Brauch, H.J.; Blankenhorn, I. Pharmaceuticals in groundwaters analytical methods and results of a monitoring program in Baden-Württemberg, Germany. J Chromatogr A 2001, 938, 199-210, doi:10.1016/s0021-9673(01)01266-3.
- Akiyama, T.; Savin, M.C. Populations of antibiotic-resistant coliform bacteria change rapidly in a wastewater effluent dominated stream. Sci Total Environ 2010, 408, 6192-6201, doi:10.1016/j.scitotenv.2010.08.055.
- Sukul, P.; Lamshöft, M.; Zühlke, S.; Spiteller, M. Sorption and desorption of sulfadiazine in soil and soil-manure systems. Chemosphere 2008, 73, 1344-1350, doi:10.1016/j.chemosphere.2008.06.066.
- Carballa, M.; Fink, G.; Omil, F.; Lema, J.M.; Ternes, T. Determination of the solid-water distribution coefficient (Kd) for pharmaceuticals, estrogens and musk fragrances in digested sludge. Water Res 2008, 42, 287-295, doi:10.1016/j.watres.2007.07.012.
- Xu, B.; Mao, D.; Luo, Y.; Xu, L. Sulfamethoxazole biodegradation and biotransformation in the water-sediment system of a natural river. Bioresour Technol 2011, 102, 7069-7076, doi:10.1016/j.biortech.2011.04.086.
- Xu, L.; Sun, X.; Ma, X. Systematic review and meta-analysis of mortality of patients infected with carbapenem-resistant Klebsiella pneumoniae. Ann Clin Microbiol Antimicrob 2017, 16, 18, doi:10.1186/s12941-017-0191-3.
- Merril, C.R.; Scholl, D.; Adhya, S.L. The prospect for bacteriophage therapy in Western medicine. Nat Rev Drug Discov 2003, 2, 489-497, doi:10.1038/nrd1111.
- shanshan, S.; Yanjun, C.; Ting, L. Distribution and drug resistance analysis of 1 326 strains of Staphylococcus aureus. Primary Medicine Forum 2022, 26, 55-57 (in Chinese), doi:10.19435/j.1672-1721.2022.35.019.
- Górski, A.; Międzybrodzki, R.; Weber-Dąbrowska, B.; Fortuna, W.; Letkiewicz, S.; Rogóż, P.; Jończyk-Matysiak, E.; Dąbrowska, K.; Majewska, J.; Borysowski, J. Phage Therapy: Combating Infections with Potential for Evolving from Merely a Treatment for Complications to Targeting Diseases. Front Microbiol 2016, 7, 1515, doi:10.3389/fmicb.2016.01515.
- Malik, D.J. Bacteriophage Encapsulation Using Spray Drying for Phage Therapy. Curr Issues Mol Biol 2021, 40, 303-316, doi:10.21775/cimb.040.303.
- Zeng, Y.; Wang, Z.; Zou, T.; Chen, J.; Li, G.; Zheng, L.; Li, S.; You, J. Bacteriophage as an Alternative to Antibiotics Promotes Growth Performance by Regulating Intestinal Inflammation, Intestinal Barrier Function and Gut Microbiota in Weaned Piglets. Front Vet Sci 2021, 8, 623899, doi:10.3389/fvets.2021.623899.
- Davies, E.V.; Winstanley, C.; Fothergill, J.L.; James, C.E. The role of temperate bacteriophages in bacterial infection. FEMS Microbiol Lett 2016, 363, fnw015, doi:10.1093/femsle/fnw015.
- Cao, F.; Wang, X.; Wang, L.; Li, Z.; Che, J.; Wang, L.; Li, X.; Cao, Z.; Zhang, J.; Jin, L.; et al. Evaluation of the efficacy of a bacteriophage in the treatment of pneumonia induced by multidrug resistance Klebsiella pneumoniae in mice. Biomed Res Int 2015, 2015, 752930, doi:10.1155/2015/752930.
- Li, C.; Jiang, M.; Khan, F.M.; Zhao, X.; Wang, G.; Zhou, W.; Li, J.; Yu, J.; Li, Y.; Wei, H.; et al. Intrinsic Antimicrobial Peptide Facilitates a New Broad-Spectrum Lysin LysP53 to Kill Acinetobacter baumannii In Vitro and in a Mouse Burn Infection Model. ACS Infect Dis 2021, 7, 3336-3344, doi:10.1021/acsinfecdis.1c00497.
- Ma, Y.; You, X.; Mai, G.; Tokuyasu, T.; Liu, C. A human gut phage catalog correlates the gut phageome with type 2 diabetes. Microbiome 2018, 6, 24, doi:10.1186/s40168-018-0410-y.
- Rasmussen, T.S.; Mentzel, C.M.J.; Kot, W.; Castro-Mejía, J.L.; Zuffa, S.; Swann, J.R.; Hansen, L.H.; Vogensen, F.K.; Hansen, A.K.; Nielsen, D.S. Faecal virome transplantation decreases symptoms of type 2 diabetes and obesity in a murine model. Gut 2020, 69, 2122-2130, doi:10.1136/gutjnl-2019-320005.
- Sharon, I.; Morowitz, M.J.; Thomas, B.C.; Costello, E.K.; Relman, D.A.; Banfield, J.F. Time series community genomics analysis reveals rapid shifts in bacterial species, strains, and phage during infant gut colonization. Genome Res 2013, 23, 111-120, doi:10.1101/gr.142315.112.
- Lim, E.S.; Zhou, Y.; Zhao, G.; Bauer, I.K.; Droit, L.; Ndao, I.M.; Warner, B.B.; Tarr, P.I.; Wang, D.; Holtz, L.R. Early life dynamics of the human gut virome and bacterial microbiome in infants. Nat Med 2015, 21, 1228-1234, doi:10.1038/nm.3950.
- Khan Mirzaei, M.; Khan, M.A.A.; Ghosh, P.; Taranu, Z.E.; Taguer, M.; Ru, J.; Chowdhury, R.; Kabir, M.M.; Deng, L.; Mondal, D.; et al. Bacteriophages Isolated from Stunted Children Can Regulate Gut Bacterial Communities in an Age-Specific Manner. Cell Host Microbe 2020, 27, 199-212.e195, doi:10.1016/j.chom.2020.01.004.
- Mirzaei, M.K.; Maurice, C.F. Ménage à trois in the human gut: interactions between host, bacteria and phages. Nat Rev Microbiol 2017, 15, 397-408, doi:10.1038/nrmicro.2017.30.
- Górski, A.; Weber-Dabrowska, B. The potential role of endogenous bacteriophages in controlling invading pathogens. Cell Mol Life Sci 2005, 62, 511-519, doi:10.1007/s00018-004-4403-6.
- Almeida, G.M.F.; Laanto, E.; Ashrafi, R.; Sundberg, L.R. Bacteriophage Adherence to Mucus Mediates Preventive Protection against Pathogenic Bacteria. mBio 2019, 10, doi:10.1128/mBio.01984-19.
- Jia, K.; Yang, N.; Zhang, X.; Cai, R.; Zhang, Y.; Tian, J.; Raza, S.H.A.; Kang, Y.; Qian, A.; Li, Y.; et al. Genomic, Morphological and Functional Characterization of Virulent Bacteriophage IME-JL8 Targeting Citrobacter freundii. Front Microbiol 2020, 11, 585261, doi:10.3389/fmicb.2020.585261.
- Yu, H.; Zhang, L.; Feng, C.; Chi, T.; Qi, Y.; Abbas Raza, S.H.; Gao, N.; Jia, K.; Zhang, Y.; Fan, R.; et al. A phage cocktail in controlling phage resistance development in multidrug resistant Aeromonas hydrophila with great therapeutic potential. Microb Pathog 2022, 162, 105374, doi:10.1016/j.micpath.2021.105374.
- Feng, C.; Jia, K.; Chi, T.; Chen, S.; Yu, H.; Zhang, L.; Haidar Abbas Raza, S.; Alshammari, A.M.; Liang, S.; Zhu, Z.; et al. Lytic Bacteriophage PZL-Ah152 as Biocontrol Measures Against Lethal Aeromonas hydrophila Without Distorting Gut Microbiota. Frontiers in Microbiology 2022, 13, doi:10.3389/fmicb.2022.898961.
- Li, M.; Chang, R.Y.K.; Lin, Y.; Morales, S.; Kutter, E.; Chan, H.K. Phage cocktail powder for Pseudomonas aeruginosa respiratory infections. Int J Pharm 2021, 596, 120200, doi:10.1016/j.ijpharm.2021.120200.
- Cui, H.; Cong, C.; Wang, L.; Li, X.; Li, J.; Yang, H.; Li, S.; Xu, Y. Protective effectiveness of feeding phage cocktails in controlling Vibrio harveyi infection of turbot Scophthalmus maximus. Aquaculture 2021, 535, 736390.
- Cong, C.; Bing-dong, W.; YuanYu-yu; Xiao-yu, L.; Li-li, W.; Ji-bin, L.; Yong-ping, X. Reviewing and Thinking on the Achievements ofBacteriophage Research in China during the 1950s. World Notes on Antibiotics 2019, 40, 401-409 (in Chinese), doi:10.13461/j.cnki.wna.005251.
- Chen, T. Trial phages to stop superbug infection - recall a long-forgotten place in the original 'Dalian Institute'. Prog in Microbiol Immunol 2010, 38, 57-61 (in Chinese), doi:10.13309/j.cnki.pmi.2010.04.018.
- Yang, Y.; Shen, W.; Zhong, Q.; Chen, Q.; He, X.; Baker, J.L.; Xiong, K.; Jin, X.; Wang, J.; Hu, F.; et al. Development of a Bacteriophage Cocktail to Constrain the Emergence of Phage-Resistant Pseudomonas aeruginosa. Front Microbiol 2020, 11, 327, doi:10.3389/fmicb.2020.00327.
- Wu, N.; Dai, J.; Guo, M.; Li, J.; Zhou, X.; Li, F.; Gao, Y.; Qu, H.; Lu, H.; Jin, J.; et al. Pre-optimized phage therapy on secondary Acinetobacter baumannii infection in four critical COVID-19 patients. Emerg Microbes Infect 2021, 10, 612-618, doi:10.1080/22221751.2021.1902754.
- Meng, F.; Sun, Y.; Yuan, G.; Gao, Y. Progress in the application of phage in the treatment of alcoholic liver disease. Chinese Hepatology 2021, 26, 956-957(in Chinese), doi:10.14000/j.cnki.issn.1008-1704.2021.09.003.
Reviewer 2 Report
This is a comprehensive review about phage therapy trials in China. However, there are many typo and grammar errors in the figs and text. They need to carfully go through the whole manuscript and correct them. And more citations are needed, such as in fig 3.
Author Response
To Editor and Reviewers:
Thank you for giving us a chance to revise the manuscript. We also thank the reviewers for constructive suggestions to help us improve the quality of the manuscript. We have made changes in the manuscript according to the reviewers’ comments and corresponding to your advice. We hope these changes make the manuscript acceptable for publication.
Response to Reviewer 2 Comments
Point 1: This is a comprehensive review about phage therapy trials in China. However, there are many typo and grammar errors in the figs and text. They need to carfully go through the whole manuscript and correct them.
Response 1: Thank you for your positive and constructive comments on our submission. We sincerely apologize for causing you this review trouble. The manuscript has been revised again by Nada Alkhorayef, Samia S. Alkhalil and Essam Eldin Abdelhady Salama who is the native English speaker. The English grammar in the manuscript has been modified. As seen from the new revised manuscript (with revisions tracked).
Point 2: And more citations are needed, such as in fig 3.
Response 2: Thank you for your comments. We have added more citations in our manuscript. As seen from the new revised manuscript (with revisions tracked).
Reviewer 3 Report
I suggest editing the introduction. It sounds a little professional, more like an introduction to a popular science article.
I suggest moving the citations to the right in Table 1. It will definitely look better.
What about the safety of using phages in therapy?
Author Response
To Editor and Reviewers:
Thank you for giving us a chance to revise the manuscript. We also thank the reviewers for constructive suggestions to help us improve the quality of the manuscript. We have made changes in the manuscript according to the reviewers’ comments and corresponding to your advice. We hope these changes make the manuscript acceptable for publication.
Response to Reviewer 3 Comments
Point 1: I suggest editing the introduction. It sounds a little professional, more like an introduction to a popular science article.
Response 1: Thank you for your review of our manuscript. We agreed with the reviewers’ proposal, and the manuscript has been changed as suggested by the reviewer. As seen from the new revised manuscript (with revisions tracked).
Point 2: I suggest moving the citations to the right in Table 1. It will definitely look better.
Response 2: Agreed. Thank you for your constructive comments for us, which really helped us improve the manuscript. And we have moved the citations to the right in Table 1. As seen from the new revised manuscript (with revisions tracked).
Point 3: What about the safety of using phages in therapy?
Response 3: Thank you for your comments. It has been proved by a large number of literature that bacteriophages are safe during application. But there are also factors that we need to consider, such as the possibility that bacteria may produce endotoxins during the lysis process. To specifically evaluate whether bacteriophages are safe, we need more data.
Round 2
Reviewer 1 Report
The authors have significantly improved the manuscript.
The authors have efficiently addressed the reviewer's comments and explained the points.
They have added a few points raised by the reviewer in the manuscript and didn't include the others.
It is indeed not necessary to include all of the points in the manuscript. The reviewer suggests the authors check all comments and questions from the previous revision again and include important points in the manuscript depending on the need.